


# Relative impacts of land use and climate change on summer precipitation in the Netherlands

Emma Daniels[1], Geert Lenderink[2], Ronald Hutjes[1], and Albert Holtslag[1]

[1]Wageningen University, Droevendaalsesteeg 3, 6708 PB Wageningen, The Netherlands
[2]KNMI, Utrechtseweg 297, 3731 GA De Bilt, The Netherlands

*Correspondence to:* Emma E. Daniels (emmadamme@gmail.com)

**Abstract.** The effects of historic and future land use on precipitation in the Netherlands are investigated on 19 summer days with similar meteorological conditions. The days are selected with a circulation type classification and a clustering procedure to obtain a homogenous set of days that is expected to favour land impacts. Changes in precipitation are investigated in relation to the present day climate and land use, and in the perspective of future climate and land use. To that end, the weather research and

forecasting (WRF) model is used with land use maps for 1900, 2000, and 2040. In addition a temperature perturbation of +1°C, assuming constant relative humidity is imposed as a surrogate climate change scenario. Decreases in precipitation are simulated following conversion of historic to present, and present to future, land use. The temperature perturbation under present land use conditions increases precipitation amounts by on average 7-8% and amplifies precipitation intensity. However, when also considering future land use, the increase is reduced to 2-6% on average, and no intensification of extreme precipitation is

simulated. In all, the simulated effects of land use changes on precipitation in summer are smaller than the effects of climate change but not negligible.

## 1  Introduction

Humans can exert influence on precipitation through modifications in land use (Mahmood et al., 2014; Kalnay and Cai, 2003) next to other anthropogenic forcings such as climate change (Zhang et al., 2007). Currently, land conversion takes place at a

rapid pace and this will likely continue in the future (Mahmood et al., 2010; Angel et al., 2011). Therefore, this type of human influence on the climate system will continue, and will probably become more significant in the coming decades (Pielke et al., 2007; Mahmood et al., 2010).

In the Netherlands, the most important land cover changes in the last century were the conversion of large heather areas into agricultural or grassland and expansion of urban areas (Feranec et al., 2010; Verburg et al., 2004). In addition, almost 1650

km2 of land was reclaimed from the sea in the former Zuiderzee, now called Lake Yssel (Hoeksema, 2007). Urban areas have increased from about 2% in 1900, to 13% in 2000, and are projected to further increase to 24% in 2040 (Dekkers et al., 2012). Precipitation in the Netherlands has increased by about 25% over the last century, especially along the West coast (Buishand et al., 2013). The increase of sea surface temperatures and changes in circulation seem to be the major causes for this increase



(Attema et al., 2014; van Haren et al., 2013). In addition, there is some evidence that urbanization plays a role (Daniels et al., 2015b).

In contrast to the above, an earlier study using a model to investigate land surface changes in the Netherlands in spring found that precipitation is in fact reduced after expansion of urban areas (Daniels et al., 2014). That study also tested the

sensitivity of precipitation to soil moisture and found a positive feedback, that is, wet (dry) soils increase (decrease) the amount of precipitation. The reduction of precipitation after urban expansion was dominated by the model's response to reduced moisture, overruling the enhanced triggering of precipitation by boundary layer processes. However, only a 4-day case study was investigated and questions can therefore be raised with respect to the climatological representability of the results. In addition, the simulated land use changes were conceptual, rather than realistic and only focused on changes in urban extent,

ignoring the expansion of agricultural areas for example.

The present study aims to improve on both aspects, by (1) sampling a larger set of meteorological cases, and (2) evaluating the effects of more realistic land cover changes. Our main interest is the precipitation response to the altered land surface and the physical processes underlying this response. We investigate this response in the summer season. The summer months typically have a larger shower activity, connected to unstable conditions. This relatively intense type of precipitation, arising

from (deep) cumulus convection, is expected to be most influenced by land surface changes (Pielke et al., 2007; Cotton and Pielke, 2007) and is typically expected to increase under climate change (Fischer et al., 2014). Also, the largest impact of urban areas on precipitation along the Dutch West coast was found in summer (Daniels et al., 2015b).

The current study also aims to put the effects of historic and future land use changes on precipitation in the perspective of climate change. This will be done by imposing an increase in overall temperatures as a surrogate climate change scenario

(Schar et al., 1996). On a global scale, climate change is expected to increase both mean and extreme precipitation in response to an intensification of the hydrological cycle (Huntington, 2006; Wu et al., 2013). Here, the precipitation response to land use changes in the Netherlands, and climate change, is investigated for multiple summer days. The selection procedure for the investigated events and the model setup will be described in the next section. Followed by the results, discussion and conclusions thereafter.

## 25 2   Data and methods

### 2.1   Case selection

Selection of days to use as case studies is conducted with the help of a circulation type classification, similar to Daniels et al. (2015a). We make use of the nine type Jenkinson-Collison Types (JCT) classification scheme. This method was developed by Jenkinson and Collison (1977) and is intended to provide an objective scheme that acceptably reproduces the subjective

Lamb weather types (Jones et al., 1993; Lamb, 1950). The classification has eight weather types (WTs) representative of the prevailing wind direction (W, NW, N, NE, E, SE, S, and SW, where W = 1 etc.) and one that is treated as unclassified (WT9). Computation of the WTs is done using 12 UTC MSLP data from ERA-Interim (Dee et al., 2011) at 16 points in the area 47.25 to 57.75°N and 3 to 12.75°E (Figure 1) with the 'cost733class' software (Philipp et al., 2010, 2014).



Previous work has shown that the downwind effects of urban areas on precipitation in the Netherlands are largest under WT9 (Daniels et al., 2015b). Under the light, unclassifiable, flow that occurs in this weather type, the atmosphere seems to be most susceptible to the land surface. All summer (JJA) days in the period 2000-2010 are classified with the JCT scheme, but only days with WT9 and more than 1 mm of precipitation at one station or more are used for further selection. The

remaining 215 days are grouped using a statistically objective k-means clustering procedure (Hartigan and Wong, 1979) in R (core team, 2013). The k-means clustering partitions $n$ observations into $k$ clusters, in which each observation belongs to the cluster with the nearest mean in a principle component space. Clustering is done to obtain a homogenous set of days with similar meteorological conditions. The similarity of the cases should result in comparable results and enable generalization of conclusions.

Seven parameters are used in the clustering procedure: (1) mean precipitation; (2) total column water; (3) vertical velocity at 700 hPa; (4) horizontal wind speed at 700 hPa; (5) K-index; (6) land-sea temperature difference; and (7) a measure of the distribution and "patchiness" of precipitation, computed as the difference between maximum precipitation and the 85th percentile. Parameters 2, 3, 4 and 5 are derived from 12 UTC ERA-Interim data averaged over the center of the Netherlands (4.75 to 5.75°E and 51.75 to 52.25°N, Figure 1). Parameter 6 is derived from ERA-Interim data as the difference between the

2 meter temperature over this land area and sea surface temperature (SST) averaged over a nearby ocean area of similar size (3-4°E and 52.25-52.75°N, Figure 1). Parameter 1 and 7 are computed over the whole of the Netherlands using daily precipitation data collected at 8 UTC from about 320 stations. The K-index (George, 1960) is a linear combination of temperature (T) and dewpoint (Td) at various levels (T850 - T500 + Td850 - (T700 - Td700)) and is a measure of convection used to forecast air mass thunderstorms. The parameter values are normalized and scaled, by subtracting the mean and dividing by the standard

deviation, before being used in the clustering algorithm.

The k-means clustering algorithm was set to use 12 clusters, repeated 1000 times and the best, stable, solution is used. A cluster with higher than mean precipitation was selected (see Figure 2), since sufficient precipitation is needed to investigate the response to alternative land use maps. Total column water is about average in the selected cluster, while it has the most negative vertical velocity (omega), of about 0.3 Pa/s. Since omega is positive with increasing pressure, this means the largest

upward speeds are selected. A large upward vertical velocity is associated with strong hourly precipitation and convective showers (Loriaux et al., 2013). Low wind speed was found to be favorable for detection of urban effects in the Netherland (Daniels et al., 2015b) and is therefore desirable. The average K-index in the selected cluster is over 20, which is the average threshold for likelihood of thunderstorms. The land-sea temperature difference is amongst the lowest. High SST is known to cause enhanced precipitation (in the coastal area) mainly in summer (Lenderink et al., 2009). This could interfere with our land

use experiments and is therefore not sought. Finally, the selected cluster has quite patchy precipitation, indicative of convective conditions as desired. The selected cluster consists of 19 days (see Figure 6 for the dates), that will be averaged on an hourly basis for many of the analyses presented in the results section.



## 2.2 Model setup

We use the non-hydrostatic Advanced Research WRF model (ARW, version 3.4.1) (Skamarock et al., 2008) on a single domain of 1000 x 1000 km (see Figure 1). The model has a horizontal grid spacing of 2.5 km and the vertical grid contains 40 sigma levels. Atmospheric and surface boundary conditions are obtained from ERA-Interim every six hours. Model output is stored
and analyzed on an hourly basis. All simulations start at 12 UTC the previous day and, unless otherwise specified, results from 0-0 are used in the analyses.

Following earlier studies with WRF in the Netherlands (e.g. Steeneveld et al., 2011; Daniels et al., 2015a; Theeuwes et al., 2013), we selected the following schemes to represent subgrid pocesses. Namely the YSU PBL scheme (Hong et al., 2006), the WRF Single-Moment 6-Class Microphysics Scheme (WSM6) (Hong and Lim, 2006), the RRTMG schemes for both long-
and shortwave radiation (Iacono et al., 2008), the Grell 3D cumulus parameterization scheme (Grell, 1993; Grell and Devenyi, 2002) and the Unified Noah Land Surface Model (Tewari et al., 2004) with the Urban Canopy Model (UCM). The UCM is a single layer model which has a simplified urban geometry. Included in the UCM are: shadowing from buildings, reflection of short and longwave radiation, wind profile in the canopy layer and multi-layer heat transfer equations for roof, wall and road surfaces (Kusaka et al., 2001; Kusaka and Kimura, 2004).

Where possible within the model domain, the European land-use map Corine (EEA, 2002) was used, supplemented with a high resolution map for the Netherlands. Corine is not available over the UK, so there the standard USGS map at 30' resolution available within WRF is used. Reclassification of the Corine land-use map is done following Pineda et al. (2004), but intertidal flats are classified as water instead of herbaceous wetlands. Three high resolution maps were used for the Netherlands: HGN1900 (Kramer et al., 2010), LGN4 (Hazeu et al., 2011; Wit, 2003; Hazeu et al., 2010), and GE2040 (Dekkers et al.,
2012), representing land use in 1900, 2000 and 2040 respectively (see Figure 3). The future map is based on the Dutch Global Economy scenario (CPB et al., 2006), a national scenario consistent with the SRES A2 scenario. The SRES scenarios have been replaced by Representative Concentration Pathways (RCP) and Shared Socioeconomic Pathways (SSP). The SRES A2 scenario is most alike SSP3 and between RCP 6.0 and 8.5 in carbon emissions. Reclassification of the Dutch land use maps is done as specified in Table 1. GE2040 unfortunately did not distinguish between different dry nature classes, so the
differentiation was copied from the LGN map. Therefore, all dry nature in GE2040 was first classified as herbaceous tundra. Next the newly classified herbaceous tundra was reclassified to Barren or sparsely vegetated areas, Evergreen needle leaf, and Deciduous broadleaf forest when it overlapped with the areas classified as such in the LGN map.

## 2.3 Model simulations

Three model simulations: HIS, REF, and FUT, are done with the land use maps of respectively 1900, 2000 and 2040 in the
Netherlands. These simulations have exactly the same boundary conditions. In 1900 the creation of land in Lake Yssel had not yet taken place. To test the effect of this conversion separately from the changes in land use an additional simulation with the historic land use map was done, this time with the current land extent (similar to that in REF). All previously non-existent land





is assumed to be covered with grassland (the most common land cover class) in this simulation. This simulation is referred to as HIS+Ys.

Furthermore, to be able to put the land cover changes in the perspective of climate change, simulations with the present and future land use maps and a temperature perturbation of +1°C are conducted. These will be referred to as REF+1 and FUT+1.

The global surface temperature is predicted to increase with at least 1°C under all concentration pathways by 2050 (IPCC, 2013). The surrogate climate change scenario is applied to the initial land and atmospheric conditions of the simulations, as well as to the driving sea surface temperature following the methodology by Attema et al. (2014), who suggest a vertically uniform temperature perturbation is appropriate at mid-latitudes. The relative humidity is unchanged in these simulations, which implies an absolute surface humidity increase of 6–7%.

Urban areas outside of the Netherlands are removed in the historic, and expanded in the future land cover scenarios, in the same way as in Daniels et al. (2014). Angel et al. (2011, 's) projections of urban land cover are used to determine the level of expansion. Across the globe, urban land cover has increased due to people migrating to urban areas and because the population density within cities decreased (Marshall, 2007). Within Europe a population density decline rate of 2% per annum has been reached between 1990 and 2000 (Angel et al., 2011). We assume a conservative increase with a decline rate of 1% for the

future. Urban areas are therefore less than doubled in our simulations, consistent with Angel's projection for Europe and Japan in 2050 with an annual density decline of 1%.

## 2.4 Precipitation data

In the Netherlands, measurements of precipitation are available from the national meteorological institute (KNMI). Gauge measurements are available on a daily basis (8-8 UTC) at about 320 stations. Gridded observations of precipitation are available

at a 2.4 km resolution on an hourly basis from (bias)corrected radar data (Overeem et al., 2009). Modelled precipitation amounts are best compared with radar data, because of the similarity in resolution and spatial extent. Unfortunately for four of the 19 selected cases there is no radar data available, so some averages shown in the results sections consist of fewer cases.

## 3 Results

The focus of this paper is on the sensitivity of precipitation to changes in land surface conditions in historical and future

perspectives. The precipitation response to the perturbations in the experiments will be described in the next section. To clarify these responses, the section after that focusses on the (differences in) atmospheric conditions and processes leading to the formation of precipitation.

In general, the WRF model overestimates precipitation amounts compared to both station and radar data (Figure 4). The days marked with red markers only have station data, and no radar data available. There is one day where precipitation amounts are

grossly overestimated, namely for June 30 2003. This day is marked with an open dot in the scatterplot. This is the only day in the selection that has easterly winds and the poor model performance could therefore be related to the chosen position of





the domain. This day was excluded from further analysis. The average wind direction on the other days is southwest, alike the year round dominant wind direction in the Netherlands.

The performance of the model to represent spatial precipitation patterns is reasonable overall, but shows quite patchy results (Figure 5). The precipitation pattern of 2000-07-29 for example is well represented by the model. This day is denoted by a triangle in Figure 4. As an example in which the model does not represent the spatial precipitation pattern well the precipitation pattern of 2007-07-22 is given. This day is denoted by a square in Figure 4. Compared to the previous example, this day is more accurately modelled in terms of amounts, but the modelled spatial distribution is quite distant from that observed. The average spatial distribution of all 18 cases overestimates the amount of precipitation compared to observed station data by almost 50%. Nevertheless, the model seems to capture the relatively high precipitation amounts in the center of the country and lower rainfall amounts in the northern parts.

The daily evolution of precipitation in observations and in the model is given in Figure 8, that will be discussed more thoroughly in the next section. Compared to radar data, the phasing of all model runs is 3 hours too early in simulating the intensification of precipitation, and the modelled precipitation peak is 2 hours too early. In addition, the average precipitation intensity is often higher than in observations. The separation of the model and observations in the evening is caused by only two days and is therefore not a generic feature. The comparison between the radar data and the modelled amounts in Figure 8 is not entirely consistent, however, since the averages are made over a different number of cases (14 vs 18 respectively). Repeating the analysis with the fewer number of cases leads to the same results.

## 3.1 Precipitation response

Despite the fact that we select days with similar atmospheric conditions, the response of precipitation to the land use and climate perturbations is not uniform and varies strongly between the different cases. In Figure 6 the relative difference of precipitation between the land cover/temperature scenarios and REF is given for each of the 19 cases. The average precipitation difference given here, is calculated over the 18 cases (excluding the 2003-06-30 case) by averaging the relative change per case. The mean precipitation difference is, on the other hand, directly calculated from the averaged precipitation amount of the 18 cases as given in Figure 7. Although the strength and sometimes the sign of the response differs between the days in every simulation, a generic picture of a decrease of precipitation appears as a response to changes in land use. From historic to present, and from present to future, land use the decreaese is about 3-5% and 2-5% respectively.

One of the averaging methods shows a difference between HIS and HIS+Ys, suggesting that the creation of land in Lake Yssel caused a moderate reduction of precipitation in the last century. The other method gives the same response for both HIS scenarios, suggesting the creation of land in Lake Yssel did not influence the total precipitation response. Either way, the model simulates a reduction of precipitation between HIS(+Ys) and REF. Similarly, the difference between FUT and REF is negative, so a reduction of precipitation is simulated by the model after incorporation of future land use.

On average, the spatial differences between the simulations are quite patchy (Figure 7). All simulations show small areas of enhancement as well as areas of reduction in precipitation. The reduction in FUT is seen over large parts of the Netherlands. Urbanization mainly takes place along the west coast, where the reduction of precipitation seems to be moderate. The relatively





small reduction might be caused by the downwind enhancement of precipitation by urban areas, though the patchiness in the rest of the country doesn't seem supportive of this hypothesis. In the HIS simulation, the largest enhancement is located on the eastern side of Lake Yssel. This increase is not visible in the HIS+Ys simulation, so it might be caused by the relatively high SST and evaporation over Lake Yssel itself and subsequent higher moisture content of the air when it reaches the coast. The

enhancement of precipitation in REF+1 and FUT+1 is most pronounced along the south-eastern border of the country. The relatively large spatial changes shown here average out to the relative changes given before in the order of 2 to 8%, which is only 0.1 to 0.6 mm. So the average changes between the runs are much smaller than the patchy spatial differences seen here.

It is interesting to see if the precipitation response to the perturbations is happening equally throughout the day, or whether it occurs during a specific moment. In the mean daily evolution of precipitation, the differences between HIS(+Ys) and REF are

hardly distinguishable (Figure 8). The differences between FUT and REF manifest themselves in the middle of the day when the intensity of precipitation is lower in FUT. This reduction of precipitation is also seen in FUT+1 and must be caused by land use changes, like the expansion of urban areas. The most pronounced temporal differences are visible in the temperature perturbation experiments: REF+1 and FUT+1. The differences are most evident in the early morning between 2 and 8 UTC. This difference is not significant as the divergence is mainly caused by the precipitation enhancement on 2000-07-05, the

day with the largest response to the temperature perturbations. So the only systematic differences between REF and other simulations are seen in FUT and FUT+1 in the middle of the day.

The surrogate climate change experiments: REF+1 and FUT+1 are conducted to allow a comparison between changes in precipitation due to land use changes and due to climate change. In our simulations, precipitation in the Netherlands increases in the temperature scenarios. The 7-8% rainfall increase in REF+1 (Figure 6) is close to the increase of about 7% $K^{-1}$ in near

surface humidity that follows from the Clausius-Clapeyron equation (O'Gorman and Muller, 2010). FUT+1 shows a more moderate increase in precipitation of 2-6%. The increase seems to be offset by the reduction in precipitation from the expected land use change that is obtained in FUT. Interestingly, it appears that the precipitation response to land use change and to the climate perturbation can be added linearly. So the mean and average values in Figure in REF+1, of respectively 8 and 7%, are reduced with the mean and average values in FUT, of -2 and -5% respectively, to attain the mean and average values in FUT+1,

of 6 and 2% respectively.

The distribution of precipitation is not well represented by the model, but consistent among the scenarios (Figure 9). The extremes of precipitation are very similar in all of the experiments, except for REF+1. The REF+1 simulation reveals a considerable increase in precipitation extremes. In the tail of the distribution the difference with REF is more than 20%. For more moderate extremes (> 15mm) the difference between REF+1 and REF is about 10%. Although mean precipitation increases in

FUT+1, the distribution remains similar to REF. Apparently extreme precipitation is in this case influenced more by land use changes than mean precipitation. The atmospheric conditions and relatively little (deep) convection in FUT+1 seem to play a role in this difference.



## 3.2 Surface and atmospheric conditions

To understand the differences between the various simulations, this section focusses on surface and atmospheric conditions. We first consider changes in the latent and sensible heat flux and changes in 2 meter relative humidity. In HIS both a higher latent and sensible heat flux are seen in comparison to REF and to HIS+Ys (Figure 10 and Table 2). This is largely caused by

the inclusion of part of Lake Yssel in the averaging, as the high lake temperature, and low albedo, causes both fluxes to be enhanced. In HIS+Ys the latent heat flux and relative humidity are somewhat higher than in REF, but the sensible heat flux is lower. Consequently the available moisture in both historical simulations will be higher and this boosts precipitation amounts. In the FUT simulations the reverse effects happens as moisture is reduced after expansion of urban areas and other land use conversions.

In REF+1 the heat fluxes are not that different from REF. Nevertheless, there is a large precipitation response. The imposed temperature perturbation with constant relative humidity increases the amount of moisture at the time of initialization and the amount that enters the model domain at the boundaries, causing precipitation to change, but fluxes to remain the same. In FUT and FUT+1 a reduction of the latent heat flux is simulated in comparison to REF. Also, in both experiments relative humidity at the surface is lower than in REF. The expansion of urban areas leads to an increase of the sensible heat flux and a decrease of

the latent heat flux, since potential evaporation is reduced within urban areas. This decreases overall moisture availability. The surface responses in FUT and FUT+1 look relatively similar, though the precipitation response relative to REF is of opposed sign in the experiments (Figure 6).

We now focus on the possibility of triggering convection by considering the atmospheric conditions. Figure 11 shows the median of the diurnal cycle of the planetary boundary layer (PBL), lifting condensation level (LCL), level of free convection

(LFC), and convective available potential energy (CAPE) calculated at the lowest model level, of the 18 cases in the REF experiment. We show the median because the mean is influenced more by outliers from individual cases. For REF+1, FUT and FUT+1 the average difference with regards to REF is given for each of these variables. The differences are normalized with respect to the mean values in REF, so a relative increase is given at every time. On average, the PBL increases to about 800 m during daytime and reaches the LCL at around 9 UTC. In the figure, the LFC remains well above the PBL and LCL. In many

individual cases, however, the LFC drops to about 800 m as well, permitting (deep) convection. The LFC reaches its lowest level at 11 UTC. This coincides with the time of the highest precipitation intensities in the model (Figure 8). CAPE increases up to 9 UTC while the LFC decreases, then stabilizes because of the rain and associated temperature and humidity changes. The early onset and intensification of precipitation in the model (Figure 8) contributes to the small buildup of CAPE and could explain the underestimation of extreme precipitation compared to observations (Figure 9). Also, there are large spatial

variations in these variables. Therefore, we computed the fraction of space and time that the PBL is higher than the LCL and LFC respectively (Table 2). We consider this a measure of the amount of triggering that occurs.

In REF+1 the temperature is higher, while the PBL is quite similar to REF. During daytime there is little difference between REF and REF+1 regarding the LCL and LFC and approximately the same amount of triggering (PBL higher than LCL/LFC) occurs (Table 2). At night the LCL and LFC are lower in REF+1 than in REF. CAPE is higher throughout the day in REF+1





than in REF, likely due to the enhanced moisture content above the PBL as a result of the imposed climate change scenario. This leads to the simulation of higher precipitation amounts and intensities in REF+1 (Figure 9). In FUT the large sensible heat flux causes the PBL to grow more during the day and stay higher during the evening than in REF. The relatively large sensible heat flux also affects and raises the LCL and LFC. In comparison to REF, CAPE decreases in FUT from 8 UTC onwards when

temperatures go up and relatively little moisture is available. Consequently, less precipitation is simulated.

In FUT+1 a combination of atmospheric processes from FUT and REF+1 can be seen. The LFC remains lower (like in REF+1), while the PBL and LCL are slightly higher (like in FUT). Accordingly, CAPE is higher than in REF in the beginning and end of the day (like in REF+1) and drops early in the day (like in FUT). In FUT+1 in total, precipitation is enhanced by the moisture availability from the boundary conditions imposed through the climate change scenario, but high intensity

precipitation is not simulated because there is little triggering and (deep) convection. Strong precipitation events are caused by convective instability, which is measured by CAPE, and generally occur during daytime. In FUT+1, CAPE is mainly enhanced during nighttime, not during daytime. The relatively low values of CAPE during daytime likely explain the absence of a response in the tail of the precipitation distribution in FUT (Figure 9).

## 4   Discussion

Although WRF is a widely used atmospheric model, questions regarding the choice of parameterization schemes and the models validity for the specific conditions always remain. The sensitivity to different parameterization schemes was not specifically investigated in this study, while this is known to be important (Gallus and Bresch, 2006; Jankov et al., 2005; Rajeevan et al., 2010; Ruiz et al., 2010; Zeng et al., 2012; ter Maat et al., 2013). The chosen YSU PBL scheme is a first-order nonlocal scheme that is widely used under convective conditions (Hu et al., 2010). The HIS, REF, and FUT experiments were duplicated without

the convection scheme, but this was found to have little effect on precipitation amounts and is therefore not shown. The utilized and presented model design is consequently only one version of reality, of which many more could be simulated. In this paper our main interest is the response of the model to changes in land use relative to climate change. Although the model's representation of precipitation is not perfect for the current climate, we believe that the current setup can still be usefull to explore the sensitivities. In addition, the model was used in a slightly different setup for a four day case in spring and comparable results

regarding the response of precipitation to increased urban areas were found (Daniels et al., 2015a). A similar reduction in precipitation was also found with the MM5 model for Europe as a whole (Trusilova et al., 2009, 2008), which gives confidence in the results.

The utilized procedure to select cases for simulation was intended to obtain a homogeneous set of days with similar meteorological conditions which were thought to favour the land surface impact on precipitation. A large spread among responses to land use and temperature scenarios was found between the cases however, so the intended comparability was not fully ac-

complished. This could be a model artefact, or a realistic response showing how differently the atmosphere reacts to similar conditions, thus showing natural variability. Nevertheless, the majority of cases has a similar sign in its response. By averaging the results we find a more representable response then the response of any single case could be. Our estimates could be biased



by the selection procedure that selected cases with rather strong convective activity. Consequently, convection will always be triggered in the selected cases and a potential feedback increasing precipitation through enhanced triggering was excluded. Examples of this feedback can be found in Findell and Eltahir (2003), Santanello et al. (2011), Taylor et al. (2012), and others. The Netherlands is however not located in a region where strong feedbacks of this type are expected (Seneviratne et al., 2006;

The GLACE Team et al., 2004) and the influence of changes in climate, SST or circulation are likely more important (Attema et al., 2014; van Haren et al., 2013). Would the selection procedure have been more successful in identifying similar events, we could have made a composite event by averaging the initial and boundary conditions, similar to Mahoney et al. (2012). Their procedure sounds promising, because it could reduce simulation time and provide a more representative response, but the selection of cases to average is apparently not straightforward.

In this study reductions in precipitation from historic to present, as well as from present to future land use are obtained for selected summer cases in the Netherlands. Observations show, however, that precipitation has on average increased by about 25% in the last century (Buishand et al., 2013). So apparently factors other than land use changes have been dominant. The observed change in precipitation was larger in the winter half year than the summer half year nonetheless, and the trend in the summer months (June – August) in the period 1951-2009 was only about 5% (Daniels et al., 2014). Hence, land surface

changes in the last century might have mitigated some of the precipitation increase in summer and hereby have contributed to the relatively low increase observed in summer. The same seems to happen in the future in the simulations: combining future land use with the expected temperature rise, reduces the precipitation increase in the model. This might only hold for summer however, because historical and theoretical evidence suggests that the precipitation response to land use changes is lower in cases with non-convective precipitation (Pielke et al., 2007; Cotton and Pielke, 2007). Studies for different types of

precipitation, taking place in other seasons, are therefore desirable as well.

The climate change scenario used here maintains constant relative humidity in the model. The resulting response in precipitation under current land cover conditions (REF+1) is close to the expected increase in near surface humidity of about 7% estimated with the Clausius-Clapeyron equation. It is interesting to note that in all simulations, except for REF+1, no differences in extreme precipitation were simulated. We note that it are not the changes in mean, but the changes in extreme

precipitation that may cause problems for society, with for example landslides or urban flooding (e.g. Feddema et al., 2005; Hibbard et al., 2010; Mahmood et al., 2014). In REF+1 precipitation over 15 mm/hr increases with 10% or more. This increase is higher than the average increase in extreme precipitation simulated by global climate models (GCMs), which is about 6% per degree global warming (Kharin et al., 2007, 2013). Mean precipitation also increases more in our simulations (7-8%) than in GCMs (3%) (Allen and Ingram, 2002). This can partly be explained because we investigate hourly data, while GCMs data is

generally daily. In addition, GCMs generally show a decrease in the occurrence frequency, and increase in intensity of precipitation. Because we only selected cases in which precipitation occurs, there can be no difference in the occurrence frequency in our simulations. Our estimates are therefore higher than those made by GCMs, but similar to comparable studies (Attema et al., 2014).



## 5   Conclusions

This paper aims to quantify the precipitation response to historic (1900) and future (2040) land use change in the Netherlands, and put this response in the perspective of climate change. To achieve this, historic, present and future land use maps are incorporated into the WRF model. In addition, simulations with a temperature perturbation of +1°C are done as a surrogate climate change scenario. The investigation is done for 18 summer days with similar characteristics that were selected with a circulation type classification and k-means clustering procedure. On average, precipitation decreases from historic to present land cover with 3-5%, and decreases with 2-5% from present to future land cover. Creation of land in Lake Yssel might have caused a decrease of precipitation, but the evidence is not exhaustive. Under the present climate, the simulated land use changes hardly have any influence on extreme precipitation.

Observations of precipitation in the last century show a year-round increase of 25%, but only 5% in summer. The results in this paper suggest that the relatively low increase of precipitation in summer due to climate change might have been offset by the effects of land use conversion. The same land use-climate compensation occurs in our simulations for the future. Precipitation increases by 7-8% on average in response to the temperature perturbation in the climate simulations and has a disproportional impact on extremes. Expected land use changes, including the expansion of urban areas, diminish this increase however. As such an average precipitation increase of 2-6% is achieved in the simulation that combines future land use with climate change. No increase in extreme precipitation is found in the combined future land use-climate change simulation. Overall, although the precipitation response to land use changes is smaller than the response to climate change, it is not negligible in the summer period in the Netherlands. Our simulations suggest this might be especially true for precipitation extremes.

*Author contributions.*   The authors collectively designed the experiments and interpreted the results. ED conducted the experiments, analysed the data and wrote the paper, that all authors commented on.

*Acknowledgements.*   ED and GL acknowledge contributions of the Knowledge for Climate programme, and RH and AH acknowledge support by the EU programme EMBRACE.



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



**Table 1.** USGS land use category descriptions and parameter settings used in WRF, with the national land use map (HGN, LGN and GE2040) classes that are reclassified as such.

| USGS land use category | Land use description | z0 (m) | Albedo (-) | green vegetation fraction (%) | Leaf Area Index | Emissivity (%) | HGN/LGN class description | GE2040 class description |
|---|---|---|---|---|---|---|---|---|
| 1 | Urban and built-up land | 0.5 | 0.15 | 0.1 | 1 | 0.88 | Buildings and roads | Urban area, commercial /industrial, seaport, building lot, infrastructure |
| 2 | Dryland cropland and pasture | 0.15 | 0.17 | 0.8 | 5.68 | 0.985 | Crops and bare soil | Arable land |
| 6 | Cropland/ woodland mosaic | 0.2 | 0.16 | 0.8 | 4 | 0.985 | Other | Recreation - single day, recreation - stay, perennial crops |
| 7 | Grassland | 0.12 | 0.19 | 0.8 | 2.9 | 0.96 | Grassland | Grassland |
| 11 | Deciduous broadleaf forest | 0.5 | 0.16 | 0.8 | 3.31 | 0.93 | Deciduous forest | Nature - dry |
| 14 | Evergreen needle leaf | 0.5 | 0.12 | 0.7 | 6.4 | 0.95 | Coniferous forest | Nature - dry |
| 16 | Water bodies | 0.0001 | 0.08 | 0 | 0.01 | 0.98 | Water | Water |
| 17 | Herbaceous wetland | 0.2 | 0.14 | 0.6 | 5.65 | 0.95 | Reed swamps | Nature - wet |
| 19 | Barren or sparsely vegetated | 0.01 | 0.38 | 0.01 | 0.75 | 0.9 | Drifting sands and sandbanks | Greenhouse horticulture, nature -dry |
| 20 | Herbaceous tundra | 0.1 | 0.15 | 0.6 | 3.35 | 0.92 | Heath land and raised bogs | Nature - dry |

**Table 2.** Mean daily (0-0 UTC) values of latent heat flux (LH), sensible heat (HFX), convective available potential energy (CAPE), precipitation (RAIN), and daytime (6-18 UTC) values of the percentage of time and area that the planetary boundary layer top is over the level of free convection (PBL > LFC), likewise for lifting condensation level (PBL > LCL), over the Netherlands for the conducted experiments

| Variable | Unit | HIS | HIS+Ys | REF | REF+1 | FUT | FUT+1 |
|---|---|---|---|---|---|---|---|
| LH | W/m$^2$ | 88.6 | 82.5 | 81.1 | 83.7 | 73.0 | 75.4 |
| HFX | W/m$^2$ | 40.2 | 38.4 | 39.4 | 38.0 | 43.8 | 42.6 |
| CAPE | J/kg | 330.1 | 311.4 | 301.2 | 360.6 | 290.1 | 346.7 |
| PBL>LCL | % | 54.2 | 54.0 | 52.7 | 52.9 | 51.0 | 51.2 |
| PBL>LFC | % | 45.3 | 45.0 | 43.7 | 44.0 | 41.7 | 42.1 |
| RAIN | mm | 7.5 | 7.3 | 7.2 | 7.7 | 6.9 | 7.5 |





**Figure 1.** Map of part of Europe showing the 16 (red) points used in the circulation type classification, the WRF model domain (black) and the land (green) and sea (blue) area used for averaging in the selection procedure.





**Figure 2.** Boxplots of the seven parameters used in the procedure to select days to simulate with the WRF model. Boxes of the days included in the selected cluster are given in orange and boxes of all summer days classified as WT 9 in the period 2000-2010 are given in white.



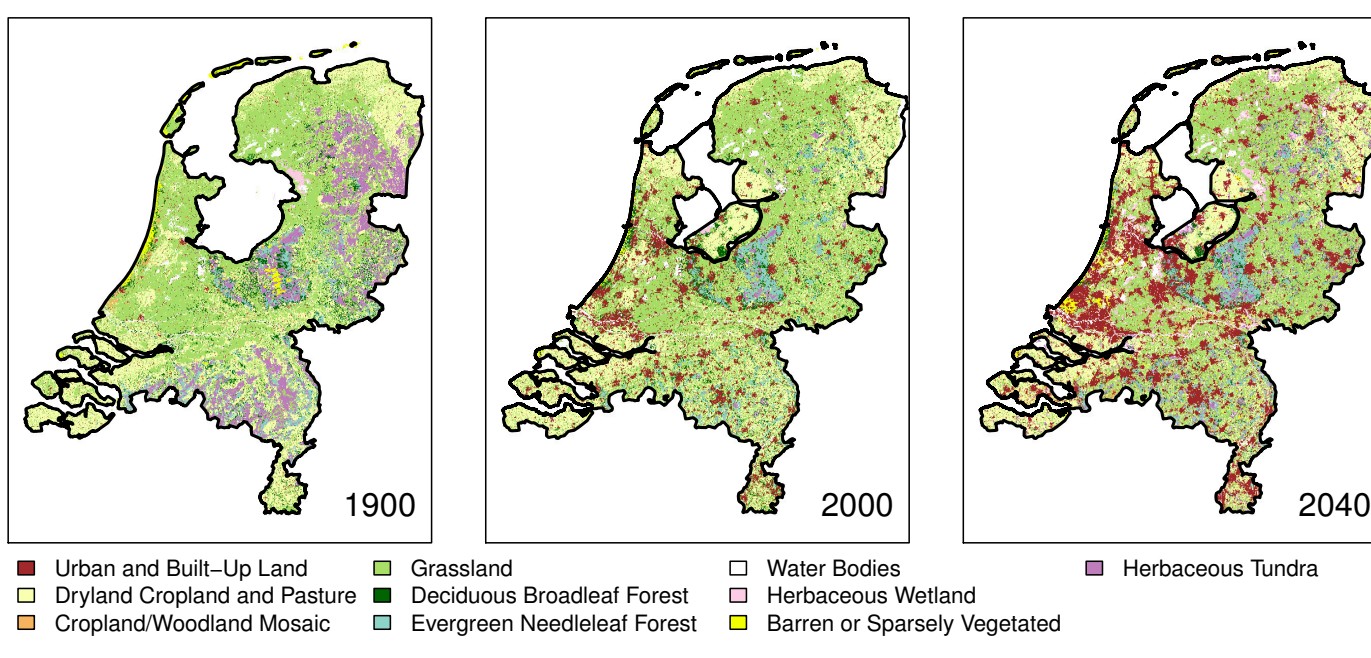

**Figure 3.** Dutch land use maps for 1900, 2000 and 2040 based on HGN1900, LGN4 and GE2040 respectively.





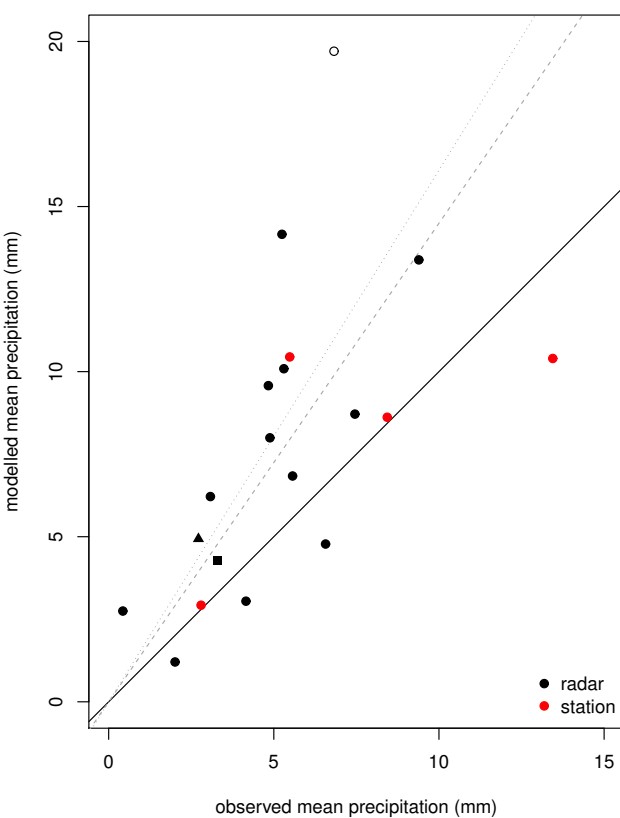

**Figure 4.** Scatterplot of observed and modelled daily mean precipitation (mm) by radar (black, 0-0 UTC) and at stations (red, 8-8 UTC) over the Netherlands. The dotted and dashed lines give a linear regression between precipitation modelled and observed by radar, respectively in- and excluding the day indicated with an open dot (2003-06-30). The days with a square (2007-07-22) and triangle (2000-07-29) are illustrated spatially in Figure 5. The solid 1:1 line represents a perfect correlation.





**Figure 5.** Daily mean precipitation (mm) simulated by the model (top) and observed (bottom) on (from left to right) 2000-07-29, 2007-07-22 (0-0 UTC) and averaged (8-8 UTC) over the 18 selected cases.





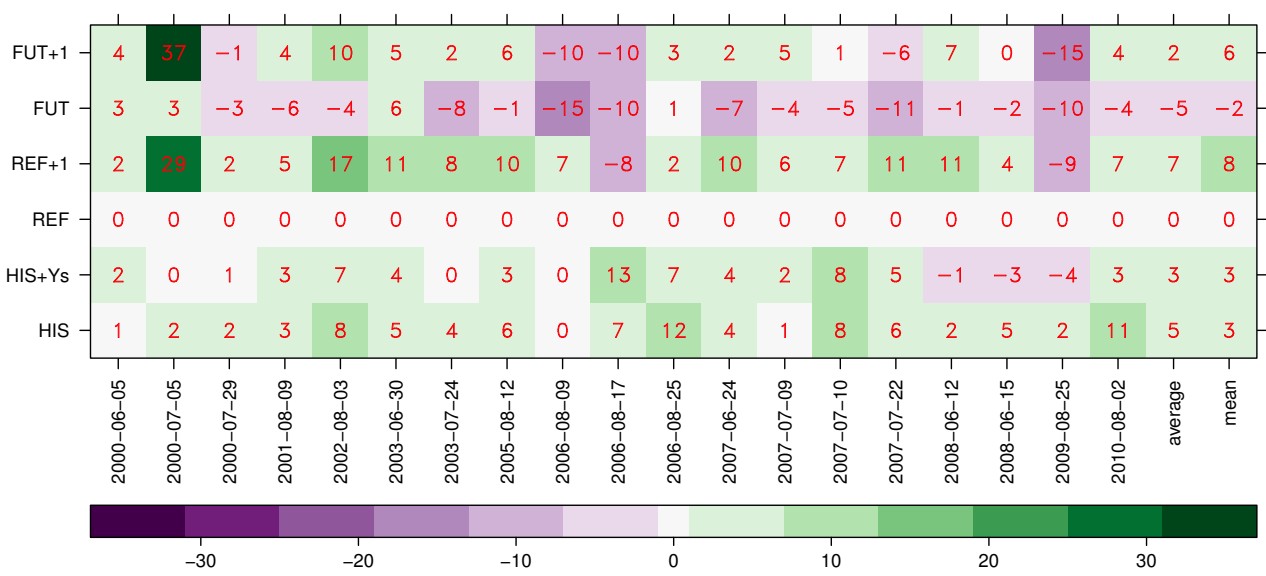

**Figure 6.** Relative precipitation difference (%) in each of the cases for all experiments compared to REF. Here the average is directly calculated over the 18 selected cases and the mean is calculated using the mean spatial differences as given in Figure 7

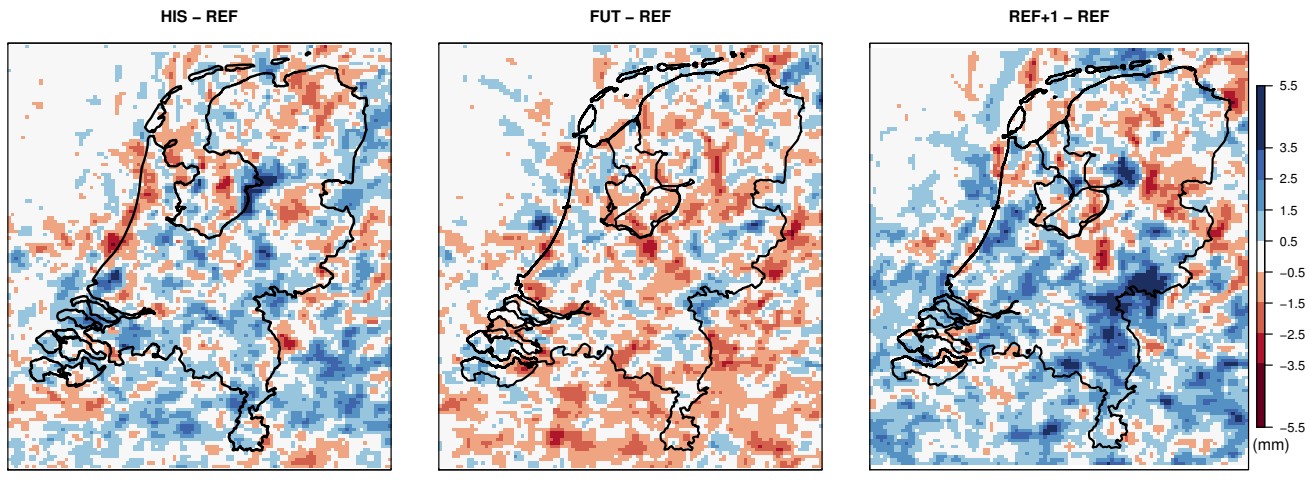

**Figure 7.** Spatial precipitation differences (mm) between the HIS, REF+1, and FUT experiments and the reference experiment.







**Figure 8.** Diurnal cycle of mean precipitation (mm) over the Netherlands in the different experiments (averaged over 18 cases) and given by radar data (averaged over 14 cases).





**Figure 9.** Distribution of hourly precipitation (mm) for each of the experiments and radar data, averaged over the 14 days that have radar data available.





**Figure 10.** Mean relative change (%) over the Netherlands in latent heat flux (LH), sensible heat flux (HFX) and relative humidity (RH) in each of the experiments in comparison to REF.







**Figure 11.** Diurnal cycle of the planetary boundary layer (PBL,solid), lifting condensation level (LCL,dashed), level of free convection (LFC,dotted) [m] and convective available potential energy (CAPE,dash-dotted) [J/kg] in the reference experiment and normalized mean difference of these variables in the experiments with a temperature perturbation and reference land cover (REF+1), future land cover (FUT), and a temperature perturbation and future land cover (FUT+1).