# Peer review of "Relative impacts of land use and climate change on summer precipitation in the Netherlands"

_Hydrology and Earth System Sciences, 2016_

## Referee Comment (RC1) · Anonymous Referee #1 · 7 Jun 2016

The submitted manuscript entitled "Relative impacts of land use and climate change on summer precipitation in the Netherlands" by Daniels et al. is a well written paper with clear clear description of methods and structure. They have analyzed the impact of land use changes on historical precipitation patterns and the impact of future land use changes on precipitation patterns. This is a very timely and relevant study. Thus it should be published. I have a few comments for improving the paper: - The results of the study show the intensifying impact of land use change on resulting precipitation patterns in the past. However, in future changes in land use have a less intensifying role of precipitation patters. The authors briefly mention the differential impacts of future land use change on extreme precipitation patterns. The paper can benefit from a deeper discussion about the impacts of future land use change on extreme precipitation patterns.

– Closely related to the previous point is the role of impervious surfaces on precipitation trends. The authors have used quite fine resolution land use maps in their study. It will be worthwhile to take into consideration the proportion and spatial patterns of increase in impervious surfaces, which will lead to greater runoff and flooding in urban areas. This measure can be accessed from fine resolution satellite images such as Landsat.

– Finally, given the main aim of the study is to examine the impact of land use changes on precipitation patterns. The authors need to take into consideration the impact of land surface temperatures in their simulations along with atmospheric conditions.

---

## Referee Comment (RC2) · Anonymous Referee #2 · 5 Jul 2016

This paper is a nice study which looks at the effects of historic and future land use on precipitation in the Netherlands. The authors do this by looking at a number of summer days with similar meteorological conditions and using WRF to examine the effect of changing land use from historic, present day and future conditions and compare this to a temperature perturbation to see which has most effect. They find that the effects of land use are not negligible although the temperature rise has a larger effect on precipitation intensity. I suggest that the paper is accepted with recourse to moderate revisions.

Suggested revisions 1. Add some detail into the abstract on what decreases in precipitation are simulated following conversion of historic to present and present to future land use. 2. It is unclear how many summer days you are examining – is it 19 or 18? Check your paper as this is not clear (both are used) 3. P5, lines 3-9: please make it

more clear how long the simulations are. This does not jump out at me from the text and I had to do some searching to find this information in the paper. Are the models just run for the 19 selected cases? 19 days for each land use scenario? If this is the case, is one model run of each day enough with which to draw strong conclusions? What role might initial conditions uncertainty play? 4. Why should you be able to add linearly the response of precipitation to land use change and climate change? This does not make intuitive sense. 5. The model inadequacy in representing current climate precipitation needs be discussed much earlier than in the discussion section – what implication does this have for the results of the study? It would be nice to see more detail in the discussion on the Trusilova et al. studies to see if the results are directly comparable. 6. It would be nice to see some discussion of natural variability and the effect of varying initial conditions on the results. 7. P10, line 10-20: Why does land use change have such a disproportionate effect in summer? 8. P10, line 28-29: The mean change simulated in this experiment is not comparable to mean change simulated by GCMs as you only simulate a few days – make this clear. In general are GCM results really comparable given that convective-permitting model experiments suggest that they do not adequately simulate summer precipitation intensities. 9. P11, line 10 onwards: could this also not be a sampling effect? Is the statement "no increase in extreme precipitation is found in the combined future land use-climate simulation" correct as this seems to contradict what you have said earlier in the paper. 10. Figure 8: Looking at the diurnal cycle of precipitation the huge difference between observed and model simulated worries me – do models give us realistic enough results to use to show sensitivity to land use change. The changes predicted by the model are much smaller than the bias between the model and observations.

Minor typos 1. P4, line 8: processes; line 23: most like 2. P5, line 14: declining rate 3. P6, line 14: in the evening is found in only two simulated days. . .; line 26: decrease 4. P8, line 16: opposite; 5. P9, line 23: useful

---

## Editor Comment (EC1) · B. Schaefli (Editor) · 13 Jul 2016

This paper has receveid to very positive reviews; both anonymous reviewers agree that the manuscript is of good to excellent scientific significance and quality. I invite the authors to respond in this public discussion to the detailed suggestions for minor revisions before preparing the revised version.
* * *

---

## Author Response (AR1)

**Relative impacts of land use and climate change on summer precipitation in the Netherlands**

Emma Daniels[1], Geert Lenderink[2], Ronald Hutjes[1], and Albert Holtslag[1]

[1]Wageningen University, Droevendaalsesteeg 3, 6708 PB Wageningen, The Netherlands
[2]KNMI, Utrechtseweg 297, 3731 GA De Bilt, The Netherlands
*Correspondence to:* Emma E. Daniels (emmadamme@gmail.com)

**1 comments from referees/public**

Anonymous Referee 1

The submitted manuscript entitled "Relative impacts of land use and climate change on summer precipitation in the Nether-

5  lands" by Daniels et al. is a well written paper with clear clear description of methods and structure. They have analyzed the impact of land use changes on historical precipitation patterns and the impact of future land use changes on precipitation patterns. This is a very timely and relevant study. Thus it should be published. I have a few comments for improving the paper: - The results of the study show the intensifying impact of land use change on resulting precipitation patterns in the past. However, in future changes in land use have a less intensifying role of precipitation patters. The authors briefly mention the

10  differential impacts of future land use change on extreme precipitation patterns. The paper can benefit from a deeper discussion about the impacts of future land use change on extreme precipitation patterns. – Closely related to the previous point is the role of impervious surfaces on precipitation trends. The authors have used quite fine resolution land use maps in their study. It will be worthwhile to take into consideration the proportion and spatial patterns of increase in impervious surfaces, which will lead to greater runoff and flooding in urban areas. This measure can be accessed from fine resolution satellite images such as

15  Landsat. – Finally, given the main aim of the study is to examine the impact of land use changes on precipitation patterns. The authors need to take into consideration the impact of land surface temperatures in their simulations along with atmospheric conditions.

Anonymous Referee 2

This paper is a nice study which looks at the effects of historic and future land use on precipitation in the Netherlands. The authors do this by looking at a number of summer days with similar meteorological conditions and using WRF to examine the effect of changing land use from historic, present day and future conditions and compare this to a temperature perturbation to see which has most effect. They find that the effects of land use are not negligible although the temperature rise has a larger

effect on precipitation intensity. I suggest that the paper is accepted with recourse to moderate revisions. Suggested revisions 1. Add some detail into the abstract on what decreases in precipitation are simulated following conversion of historic to present and present to future land use. 2. It is unclear how many summer days you are examining – is it 19 or 18? Check your paper as this is not clear (both are used) 3. P5, lines 3-9: please make it more clear how long the simulations are. This does not jump out at me from the text and I had to do some searching to find this information in the paper. Are the models just run for the 19 selected cases? 19 days for each land use scenario? If this is the case, is one model run of each day enough with which to draw strong conclusions? What role might initial conditions uncertainty play? 4. Why should you be able to add linearly the response of precipitation to land use change and climate change? This does not make intuitive sense. 5. The model inadequacy in representing current climate precipitation needs be discussed much earlier than in the discussion section – what implication does this have for the results of the study? It would be nice to see more detail in the discussion on the Trusilova et al. studies to see if the results are directly comparable. 6. It would be nice to see some discussion of natural variability and the effect of varying initial conditions on the results. 7. P10, line 10-20: Why does land use change have such a disproportionate effect in summer? 8. P10, line 28-29: The mean change simulated in this experiment is not comparable to mean change simulated by GCMs as you only simulate a few days – make this clear. In general are GCM results really comparable given that convective-permitting model experiments suggest that they do not adequately simulate summer precipitation intensities. 9. P11, line 10 onwards: could this also not be a sampling effect? Is the statement "no increase in extreme precipitation is found in the combined future land use-climate simulation" correct as this seems to contradict what you have said earlier in the paper. 10. Figure 8: Looking at the diurnal cycle of precipitation the huge difference between observed and model simulated worries me – do models give us realistic enough results to use to show sensitivity to land use change. The changes predicted by the model are much smaller than the bias between the model and observations. Minor typos 1. P4, line 8: processes; line 23: most like 2. P5, line 14: declining rate 3. P6, line 14: in the evening is found in only two simulated days: : :; line 26: decrease 4. P8, line 16: opposite; 5. P9, line 23: useful

**2 author's response**

Author comment to reviewer 1

Thanks for your timely and positive comments. Regarding the first comment about extreme precipitation patterns, we foresee expanding the last paragraph of section 3.1 and/or adding a figure with e.g. the 95th percentile distribution of precipitation for the REF, REF+1 and FUT+1 case. For the second comment, regarding the role of impervious surfaces, accessing this on a high(er) resolution for the current study is somewhat out of scope. Impervious surfaces are however taken into account implicitly by WRF in the Urban Canopy Model and its parametrizations. From the results of this study and a previous study, the role of impervious surfaces seems to be substantial because they alter the surface water balance and related sensible heat flux, so this is certainly something to keep in mind in future work. Finally, land surface temperatures are simulated vey accurately by the model (much better than precipitation) so this has already been taken into account. In an earlier paper - Daniels

et al 2015a, referred to in the reference list – this has been shown and discussed in detail, also in relation to the urban heat island.

Author comment to reviewer 2

5   Thanks for your positive words and suggestions for improvement. Regarding your suggested revisions: 1. We will add the 3-5% increase resulting from conversion of historic to present and 2-5% reduction following present to future land use to the abstract. 2. 19 cases were initially selected, but only 18 were used in the analysis. We will check the paper for clarity on this. 3. The model is run for 48 hours, including 12 hours of spin-up from 12 to 00 UTC, 24 hours of simulation and 12 additional hours to be able to compare to both radar data (00-00UTC) and station data (8-8UTC). We will make this more clear. The

10  model is indeed run for 19 cases for each land-use/temperature perturbation. Although 19 cases might still not be enough, it is substantially more than a single case study. In addition, we thought the selection procedure for the cases would assist in drawing relatively strong conclusions, but the results were more heterogeneous than hoped for. The initial conditions might have affect the results, though a sensitivity analysis was performed for some of the cases by starting the runs up to 3 hours earlier or later and this had relatively effect and WRF seems pretty robust in its predictions. 4. The linearity in adding the

15  responses is indeed not intuitive, though something we have encountered with WRF before. In Daniels et al. 2015a -referred to in the reference lista similar linearity is found for changes in the latent heat flux and related parameters. We are unclear on the implications, but thought it is worth drawing the attention of the reader to. 5. The model inadequacy in representing the current climate precipitation can be discussed in the beginning of the results section. What the implications are for the study is unclear. WRF is a commonly used model that does not do worse on predictions than similar mesoscale or alike climate models. The

20  usefulness of alike studies can be sought in understanding the processes governing the changes -in precipitation in our case- not in the numeric outcomes of the models per se. We hope to further such understanding through section 3.2 explaining the atmospheric processes involved. More in-depth discussion on the Trusilova et al. studies will be added. 6.We can add some discussion on natural variability and initial conditions. From previous work, the soil moisture initialisation seems to be one of the most important in general. In the Netherlands those conditions are generally at field capacity however due to the frequent

25  rain and high ground water table and can therefore be expected to have limited influence. 7. Land use change is considered to have a larger effect on convective precipitation, which mostly occurs in summer. 8. We will make this distinction more clear. Neither GCMs nor WRF are unfortunately able to adequately simulate precipitation intensities. 9. This could very well be a sampling effect. The sentence on extreme precipitation is certainly correct. In this case the FUT+1 case is meant, where land use changes seem to counter the increase in extreme precipitation that is observed in REF+1. 10. This comment touches upon

30  our main concern while preparing the paper. The analysis was repeated twice for a different set of data, but led to similar results. This gives us confidence in the response of the model, but whether this response is realistic cannot be inferred. Nevertheless, we hope to add to the knowledge on land atmosphere processes by describing the surface and atmospheric conditions and processes in detail. Sharing successful and less successful methodologies – in this case the selection procedure (clustering approach)- is in our opinion at least as important as the numeric conclusions that can be drawn from the simulations, which

35  should be treated with caution due to sampling and model issues.

**3   author's changes in manuscript**

[revised manuscript text omitted]